# Comparative Performance Testing of Respirator versus Surgical Mask Using a Water Droplet Spray Model

**DOI:** 10.3390/ijerph18041599

**Published:** 2021-02-08

**Authors:** Paul T. J. Scheepers, Heiman F. L. Wertheim, Maurice van Dael, Rob Anzion, Henk Jan Holterman, Steven Teerenstra, Martijn de Groot, Andreas Voss, Joost Hopman

**Affiliations:** 1Department for Health Evidence, Research Laboratory Molecular Epidemiology, 6500 HB Radboudumc, Nijmegen, The Netherlands; maurice.vandael@radboudumc.nl (M.v.D.); rob.anzion@radboudumc.nl (R.A.); 2Department of Medical Microbiology, Radboudumc, 6500 HB Nijmegen, The Netherlands; heiman.wertheim@radboudumc.nl (H.F.L.W.); Andreas.voss@radboudumc.nl (A.V.); joost.hopman@radboudumc.nl (J.H.); 3Radboudumc Centre for Infectious Diseases, Radboudumc, 6500 HB Nijmegen, The Netherlands; 4Wageningen Plant Research, Wageningen University and Research, 6700 AA Wageningen, The Netherlands; henkjan.holterman@wur.nl; 5Department for Health Evidence, Section Biostatistics, Radboudumc, 6500 HB Nijmegen, The Netherlands; Steven.Teerenstra@radboudumc.nl; 6Radboudumc REshape Center, Radboudumc, 6500 HB Nijmegen, The Netherlands; Martijn.degroot@radboudumc.nl; 7Department of Medical Microbiology and Infectious Disease, Canisius Wilhelmina Hospital, 6532 SZ Nijmegen, The Netherlands

**Keywords:** respiratory protective equipment, virus transmission, aerosols, particle size distribution, COVID-19 pandemic

## Abstract

Background. During the SARS-CoV-2 pandemic, there was shortage of the standard respiratory protective equipment (RPE). The aim of this study was to develop a procedure to test the performance of alternative RPEs used in the care of COVID-19 patients. Methods. A laboratory-based test was developed to compare RPEs by total inward leakage (TIL). We used a crossflow nebulizer to produce a jet spray of 1–100 µm water droplets with a fluorescent marker. The RPEs were placed on a dummy head and sprayed at distances of 30 and 60 cm. The outcome was determined as the recovery of the fluorescent marker on a membrane filter placed on the mouth of the dummy head. Results. At 30 cm, a type IIR surgical mask gave a 17.7% lower TIL compared with an FFP2 respirator. At 60 cm, this difference was similar, with a 21.7% lower TIL for the surgical mask compared to the respirator. When adding a face shield, the TIL at 30 cm was further reduced by 9.5% for the respirator and 16.6% in the case of the surgical mask. Conclusions. A safe, fast and very sensitive test method was developed to assess the effectiveness of RPE by comparison under controlled conditions.

## 1. Introduction

The method of transmission of SARS-CoV-2 has been the subject of much discussion since the virus was first reported. The WHO advice on SARS-CoV-2 infection prevention and control for healthcare workers is based on transmission through respiratory droplets and contact and assumes the possibility of airborne transmission mainly due to aerosol-generating procedures [1]. Recent research has demonstrated the presence of SARS-CoV-2 RNA in the air in COVID-19 patient wards and ICUs [2,3,4], and experimental studies showed that SARS-CoV-2 remained viable in artificially created aerosols for at least 3–16 h [5,6]. Concluding evidence from animal or human studies to support a risk-based calculation of aerogenic SARS-CoV-2 infection is still missing [7].

According to the WHO, a surgical mask provides adequate protection against inhaling respiratory droplets [1]. The use of type FFP2 or N95 respirators is considered good practice in areas where aerosol-generating procedures in COVID-19 (suspected) patients are performed [8].

FFP2 is short for “filtering facepiece particles” and was developed in the European Union as a general-purpose respirator, with the number 2 indicating the degree of protection from hazardous dust and fume particles [9]. FFP2 and N95 respirators offer similar levels of protection, as determined by the Centers of Disease Control and Prevention [10]. The choice of respirators as the preferred protective equipment is based on laboratory tests in accordance with international standards (Table 1). Sodium chloride (NaCl) 2% in water is used to create “solid particles” by atomization using a nebulizer and paraffin fumes to create an oily mist. Both tests are mainly representative of emissions of toxic substances but not of the daily clinical practice in which healthcare workers are exposed to emissions generated during the care of COVID-19 patients. The test substances have different properties than the bioaerosols that are released in clinical settings, for which different requirements for the protective equipment are defined. The surgical mask does not normally undergo testing based on the principle of protecting the person wearing the mask from external hazards. The type IIR surgical mask is used by professional healthcare workers and adheres to extra requirements (compared to type I and II) including splash resistance. Additionally, when used as a respiratory device for personal protection, the performance requirement for FFP respirators of EN 149:2001 + A1:2009 applies [9]. Notwithstanding the original purpose of both masks, evidence that a surgical type IIR mask provides equal protection against viral pathogens as compared to respirators is increasing. Two recent systematic reviews showed that healthcare workers wearing a surgical mask have a similar probability of contracting influenza as compared to healthcare workers wearing a N95 respirator [11,12].

For protection against the smaller aerosols that can be generated during certain procedures such as suction, higher demands are placed on the filter efficiency that must go hand in hand with relatively good breathing ability. This only works with high-quality filter material incorporated in a design with a good fit to achieve the required protection. In view of the many questions and justified concerns, we tested by comparison the performance of a FFP2 respirator and surgical mask with droplets of a wide size range using an experimental set-up that was developed in an early phase of the COVID-19 pandemic.

## 2. Materials and Methods

The experimental conditions are described below. First, the spray characteristics of the used nebulizer are described with a focus on the volume and number size distribution characteristics of the sprayed water droplets. Next, the procedure for the total inward leakage (TIL) testing of the respirators and surgical mask is described. TIL is defined as the combination face seal leakage and possible penetration through the filter [9].

### 2.1. Test Set-Up

We used a custom-built cross-flow nebulizer with an open outlet to generate a spray of droplets of pure (MilliQ) water with a 1.0 mg/L solution of a non-volatile fluorescent marker, fluorescein in pure water. With an airflow of 13 L/min, we generated a conical jet spray that was pointed to the nose of an anatomically correct dummy head, the Sheffield head described in the European standard for testing FFP respirators [9]. The particle size distribution of the jet spray was measured by phase-doppler anemometry using a Phase Doppler Particle Analyzer (PDPA) system (TSI, Shoreview, MN, USA). The PDPA was set up to measure droplets in the range 0.8–300 μm. The nebulizer and the measurement set up can be found in Appendix A. Table 2 shows the size distribution by volume, and Figure 1 provides the number-based size distribution for the water droplet spray measurements at 30 and 60 cm. Cumulative size distributions for both number and volume are provided in Appendix A.

### 2.2. TIL Testing Procedure

The performance of the surgical masks and FFP2 respirators was tested at two distances (Figure 2), 30 and 60 cm, considered to be relevant distances in the workplace when taking care of COVID-19 patients by local nurses and doctors. At 30 cm, the spraying was done for 10 s. At 60 cm, the spraying was done at double load in two periods of 10 s each with a 5 min interval, which resulted in a sprayed liquid volume of approximately 1.0 mL. The total sampling duration was set at 15 min for both spray distances. Additionally, RPEs were tested at 30 and 60 cm, both with and without a face shield (PIANT, Waalwijk, The Netherlands).

This laboratory-based set-up was used to compare the surgical IIR mask (Medicon, Montreal, Canada) with 3M’s FFP2 1862+ Aura respirator (near equivalent to N95). To determine the TIL, the water droplets were collected on a binder-free quartz microfiber membrane filter Whatman GF/C (Sigma Aldrich, Zwijndrecht, the Netherlands) that was placed in the mouth opening of the dummy head (see Figure 2b). The RPEs to be tested were placed on the dummy head according to instructions provided by the supplier. Quantification of TIL was achieved by analysis of the amount of fluorescein captured on the membrane filter at an airflow of 25 L/min, which corresponds to light exercise. The membrane filters were extracted by 15 min of mechanical shaking with MilliQ water followed by removal of any solids by centrifugation at 3220 rfc for 5 min. For detection of fluorescein, the excitation wavelength was set at λex 460 nm and the wavelength for emission was set at λem 515 nm. Detection was performed using a fluorescence spectrophotometer (SpectraMax iD5 Multi-mode microplate reader, Molecular Devices) in the polarization mode. Small aliquots of the test solutions were measured in 200 μL wells of a 96-well plate together with a series of calibration standards consisting of known quantities of the fluorescein that were applied on the membrane filter surface (0, 5, 50, 100, 250, 500, 1000, 2000, and 5000 ng). With this approach, we adjusted for the background signal generated by the filter surface and also for the recovery of fluorescein during the sample pretreatment. On every day RPEs were tested, we prepared a new calibration curve. The mean R^2^ corresponds to 0.98 with variability in the slope factor of 3.2%, calculated as relative standard deviation based on four calibration curves. Filter and solution blanks and the calibration standards were analysed in the same run (on the same 96 wells plate). The limit of quantification was 50 ng/filter and the coefficient of variance of the analysis was 11%. All comparisons between surgical masks and respirators were based on test runs performed on the same day.

### 2.3. Statistical Analysis

The TIL reduction percentages were calculated as (C_0_ − C_1_)/C_0_) × 100% with C_0_ and C_1_ as the concentration of retrieved fluorescein (ng) at the reference and intervention conditions, respectively. Distributions of the TIL in each condition were assessed using box-plots and described using descriptive statistics (mean and standard deviation, or median and quartiles if skewed). Differences in the means between conditions were tested, accounting for possible different variances across the conditions. To this end, the TIL data were analysed using a linear mixed model with conditions as fixed effects and a heteroscedastic covariance matrix of the residuals, blocked by condition. For the distance of 30 cm, multiple testing of differences was corrected for using a Bonferroni correction. The comparisons of a priori interest were FFP2 versus surgical mask, FFP2 with and without face shield, and surgical mask with and without face shield, and so a two-sided *p*-value of 0.05/3 = 0.016 was used for statistical significance. For the distance of 60 cm, only the comparison between FFP2 and surgical mask was of primary interest, and no correction for multiple testing was applied, so *p* = 0.05 was used for statistical significance.

## 3. Results

In our comparative measurements, the surgical mask gave a 17.7% lower TIL as compared to the FFP2 respirator in the 30 cm condition (Figure 3 and Appendix A). The *p*-value for this comparison was 0.03, but was not statistically significant after correction for multiple testing. At 60 cm, the TIL was 21.7% lower for the surgical mask compared to the FFP2 respirator (*p* = 0.004, statistically significant). When adding the face shield as extra protective device at 30 cm, the TIL was further reduced by 9.5% for the FFP2 respirator and 16.6% in the case of the surgical mask. When comparing the ‘face shield only’ condition as a reference at 60 cm and adding RPE, the reductions were higher: 16.1% and 36.3% for FFP2 respirator and surgical mask, respectively (Figure 3b). The latter comparison had a nominal *p*-value of 0.01.

## 4. Discussion

We found additional evidence to support the use of a type IIR surgical mask as an alternative to the FFP2 respirator for protection during COVID-19 patient care in the absence of aerosol generating procedures. Assuming the spread is mainly through droplets, these outcomes are in line with the findings of two recent systematic reviews not reporting differences in confirmed influenza virus infections in healthcare workers wearing a N95 respirator compared to staff using a surgical mask [13,14]. Our results confirm previous results that suggest that healthcare workers who come close to COVID-19 patients can be adequately protected by simple measures, such as a surgical mask combined with a face shield or other eye protection. We recommend the use of respirators of FPP2 quality or equivalent during aerosol-generating procedures as the fine respirable fraction of droplets should be captured in a filter consisting of several layers with high-quality filter technology.

The observed variability in our results reflects both measurement variability and genuine variability. Placing the mask and aligning the nebulizer to the dummy head may have contributed to measurement variability. Variability in the spray load was controlled by correcting the result by the sprayed liquid volume (determined by gravimetry). Important contributing factors to genuine variability were sample-to-sample differences in the test items and fit of the tested masks on the dummy head. In reality, the way face masks are placed on the face would likely result in an even larger variability than what we reported based on the laboratory tests with the dummy head.

We suggest that the increase in the TIL variability of combining the RPE with a face shield compared to the test results of the RPE only observed at 30 cm (Figure 3a) is attributable to turbulence of the air flow around the edges of the face shield. This explanation is consistent with the observation of a much larger spread of the observations in the ‘face shield only’ conditions compared to the surgical mask and FFP2 respirators tested without a face shield. Our data analysis, however, did not indicate any influence of the face shield on the TIL performance of the RPE (no interaction observed). Differences in TIL between the 30 and 60 cm test conditions are small. To avoid a too-low recovery on the membrane filter, we doubled the spray load for the 60 cm test condition. This two-fold increase in the source strength also contributed to the difference observed between the 30 and 60 cm test conditions. As can be seen in Figure 3, the TIL is higher for the 60 cm condition compared to the 30 cm condition. As the concentration decreases exponentially with distance, the covered fluorescein would be expected to be lower at 60 cm compared to 30 cm even when taking into account the twofold increase in load. Finding similar or even higher TIL and a much smaller variance indicates that small droplets (that remain airborne at 60 cm) have a relatively higher contribution to the overall TIL performance as compared to the 30 cm condition. There are two reasons why smaller particles are more abundant: large droplets did not reach the 60 cm point because of gravity, and at 60 cm, the droplets have lost more of their size due to water evaporation. Despite the smaller droplet size, the amount of fluorescein remains the same. Smaller droplets follow the air flow more easily, resulting in a higher contribution of face seal leakage. This also explains that the face shield only solution results in a higher TIL compared to RPEs. In short, the face shield is effective to prevent contamination by splashes and large droplets that impact on the shield due to inertia, but cannot be expected perform as well as RPE.

The strength of our approach is the use of a wide size range of droplets, reflecting the risk of exposure to droplets potentially loaded with virions as opposed to the standards that have a focus on particle counts (often referred to as ‘aerosols’), for which the risk of infection is not as yet confirmed [8]. In a clinical setting, exposure to a few single large particles could represent a risk if carrying a high virus load [3]. The use of the fluorescent marker allows a very high sensitivity of the test procedure and the fluorescent marker concentration reflects the much higher virus load of larger droplets. This method of quantification is much more relevant to the COVID-19 situation than the particle counts that are often used to report the results of performance tests of RPE, i.e., in the European standards [5,6,7]. These existing FFP efficiency tests with NaCl and paraffin are not representative for hospital practice with infectious bioaerosols. Both tests are mainly representative of emissions of toxic particles, not of the daily clinical practice in which healthcare workers are exposed to droplet emissions related to the care of COVID-19 patients. The test substances have different properties than the bioaerosols that are released in clinical settings, for which different requirements for the protective equipment are defined [9,10,12].

Our test method with the Sheffield head and fluorescent water droplet spray provides an alternative and more relevant test condition that comes closer to clinical practice. Limitations of the test procedure include the simple approach with regard to the direction of the airflow (inward only). It would be useful to consider adding a spray condition at a different angle, e.g., from the side or from below: it is possible that for surgical masks, duct-shaped pleats formed at the side of the head would enhance the contribution of face-seal leakage. In addition, in many laboratory-based tests, the conditions are fixed, and thus do not reflect the wide range of conditions of wearing RPE encountered in real life.

This laboratory set up is different compared to tests that are used for pre-market certification of FFP respirators (EN149:2009). This test may serve useful as additional ‘post-market’ evaluation by the procurement department of healthcare facilities [15]. The test set up could be further optimized. One improvement would be to add the possibility of more accurate quantification of the spray load. This would allow determination of a measurement-verified protection factor.

## 5. Conclusions

With the use of a fluorescein solution in pure water, a safe, fast, and very sensitive test method is available to assess the effectiveness of respiratory protective and face shield equipment by comparison under controlled conditions. The test makes it possible to compare TIL for different types of wearable barriers in a standardized way. The results of our study suggest that surgical IIR masks have the ability to protect healthcare workers taking care of COVID-19 patients in the absence of aerosol generating procedures.

## Figures and Tables

**Figure 1 ijerph-18-01599-f001:**
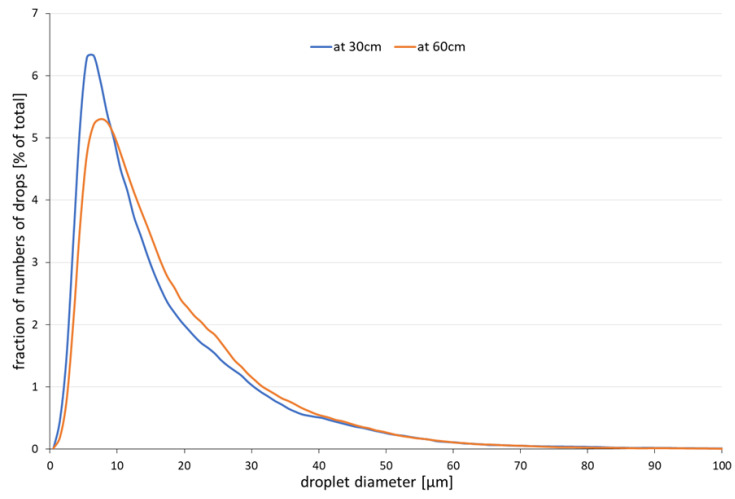
Particle size number distribution in the spray of water droplets at 30 and 60 cm.

**Figure 2 ijerph-18-01599-f002:**
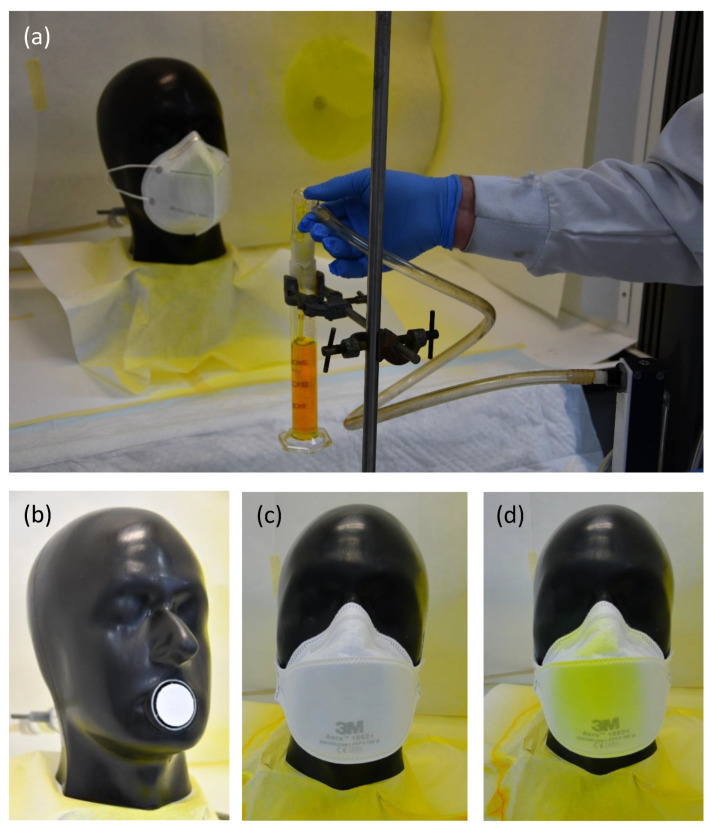
Test set-up for total inward leakage (TIL) test at 30 cm distance (**a**); Sheffield head with membrane filter placed in the mouth opening (**b**); 3M FFP2 Aura mounted on dummy head before (**c**) and after (**d**) spray sequence (with the yellow stain of fluorescein showing soiling of the respirator).

**Figure 3 ijerph-18-01599-f003:**
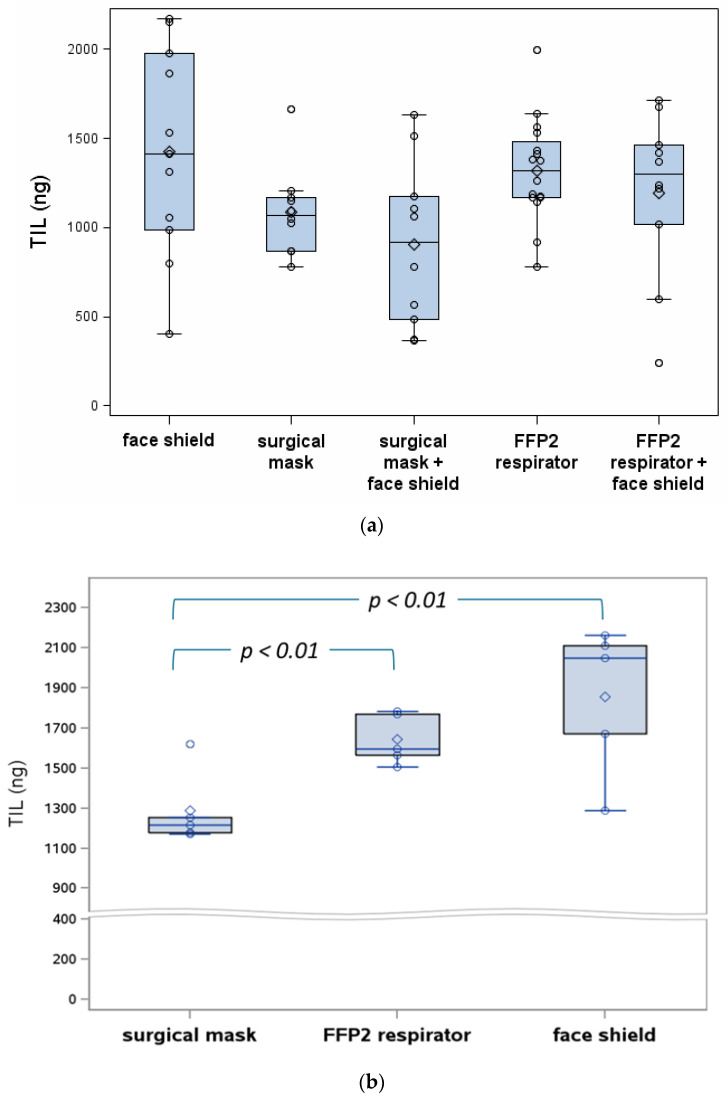
TIL expressed as recovered fluorescein at 30 cm (*N* = 10–16) (**a**) and 60 cm (*N* = 5) (**b**). The diamonds correspond to the means.

**Table 1 ijerph-18-01599-t001:** Test conditions and performance standards for respirators and surgical masks in Europe and the US.

Type	European Standard	Testing Substance	Particle Size (µm)	Filter Penetration	Bacterial Filtration Efficiency
N95	NIOSH 42 CFR Part 84 [10]	NaCl	0.3	<5%	-
FFP2	EN 13274-7: 2019 [13]	NaCl	0.6	<6%	-
		Paraffine	0.3	<6%	-
Surgical mask (IIR) ^1^	EN 14683: 2019 [14]	*Staphylococcus aureus*	3.0 ± 0.3	-	≥98%

^1^ When used as personal protective equipment, in addition the requirements for respirators in EN149 apply.

**Table 2 ijerph-18-01599-t002:** Droplet size characteristics of water droplets spray by volume of the jet spray (mean and standard deviation). More details are presented in Appendix A.

Parameter	Particle Size Distribution by Volume (µm)	
	30 cm (*N* = 4)	60 cm (*N* = 4)
Minimum	1.2 ± 0.2	1.0 ± 0.1
10th percentile (P10)	29 ± 1.3	25 ± 0.5
Median (P50)	76 ± 12	53 ± 2.1
90th percentile (P90)	226 ± 40	167 ± 52

## Data Availability

Not applicable.

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
