# Peer review of "Comparative Performance Testing of Respirator versus Surgical Mask Using a Water Droplet Spray Model"

_ijerph, 2021, doi:10.3390/ijerph18041599_

Round 1

Reviewer 1 Report

The manuscript title “Comparative performance testing of respirator versus surgical mask using a water droplet spray model” is very interesting while it needs some improvement. These are the comments:

Comments:

  • Abbreviations should be defined in parentheses the first time they appear in the main text/figure.
  • The two different abbreviations were used for COVID-19, SARS-COV-2 and COVID-19.
  • All the figures need to be elaborated accordingly in the legend section.
  • In figure 3, the lower panel the mean values do not match with the upper panel for the surgical mask, FFP2 respirator only and, with the data provided in supplementary Table-S1. And there is no need to split the Y-axis of the lower graph, because there is no data in bottom half of graph.
  • Authors has mentioned about CDC, which CD needs to be defined appropriately.
  • In so many places, the word RPE solution is used, which is confusing here. Need to replace with other words.
  • Arbitrary use of names, for surgical mask different names, were used viz. RPE, RPE solution, surgical mask, IIR mask and SM. Which needs to be taken care.
  • In the discussion section (line 203-207), author has given an explanation of relatively high TIL at 60 cm is “smaller droplets follow the air flow more easily, resulting in a higher contribution of face seal leakage”. While at given time and condition at both 30 and 60 cm distance, the dummy head with membrane filter was provided with equal exposure time of 15 minutes. So, at 30 cm distance the dummy head had exposure to both small and large aerosol/droplets and should have more TIL, while it is not like so. The higher TIL at 60 cm might be because of doubling the aerosol generation challenge compare to 30 cm distance i.e twice as compared to 30 cm.

Author Response

The authors much appreciate the very useful suggestions for clarification and improvement of the manuscript. Below we have responded point-by-point and also suggested some changes to resolve the concerns of the reviewers.

Reviewer-1

Abbreviations should be defined in parentheses the first time they appear in the main text/figure.

The two different abbreviations were used for COVID-19, SARS-COV-2 and COVID-19.

Response: we have introduced all abbreviations upon first use and have also corrected the spelling of some of the abbreviations. We have strictly adhered to ‘SARS-CoV-2 (and not adopted the suggested ‘SARS-COV-2’) as the most common and accepted spelling of the name of virus and COVID-19 as the disease.

All the figures need to be elaborated accordingly in the legend section.

Response: we are not sure if we understand the reviewer’s suggestion as the submission uses the standard format used for publication with placement figures and their legends in the main text.

In figure 3, the lower panel the mean values do not match with the upper panel for the surgical mask, FFP2 respirator only and, with the data provided in supplementary Table-S1. And there is no need to split the Y-axis of the lower graph, because there is no data in bottom half of graph.

Response: upper panel (a) presents the results at 30 cm and in the lower panel (b) the results of testing at 60 cm are presented. These results are different because of the difference in test conditions indicated in the Figure legend as ’30 cm’ and ‘60 cm’). You will find a more precise description of these test conditions in the methods section (L120-127). To clarify the different outcome of the test depending on the test condition we added an explanation to the discussion on how the test conditions impact the study findings (see below and 235-246). The data in Table S1 correspond to the diamonds in the boxplots, and these check out. To avoid confusion, we added in the legend that the diamonds correspond to the means. If the y-axis was not split, the box plots would be squeezed and differences would be hard to see.

Authors has mentioned about CDC, which CD needs to be defined appropriately.

Response: We added the full name of the CDC in L554 of the revised text and used that also as the entry for the reference list.

In so many places, the word RPE solution is used, which is confusing here. Need to replace with other words. Arbitrary use of names, for surgical mask different names, were used viz. RPE, RPE solution, surgical mask, IIR mask and SM. Which needs to be taken care.

Response: the reviewer is right that there are many names for the class of protective equipment and this use of not harmonized globally. We consider term respiratory protective equipment (RPE) a useful term to as subclass of all personal protective equipment (which also includes, gloves, glasses, garments, earplugs). We agree that ‘RPE solutions’ is less suitable replaced this by ‘RPEs’

In the discussion section (line 203-207), author has given an explanation of relatively high TIL at 60 cm is “smaller droplets follow the air flow more easily, resulting in a higher contribution of face seal leakage”. While at given time and condition at both 30 and 60 cm distance, the dummy head with membrane filter was provided with equal exposure time of 15 minutes. So, at 30 cm distance the dummy head had exposure to both small and large aerosol/droplets and should have more TIL, while it is not like so. The higher TIL at 60 cm might be because of doubling the aerosol generation challenge compare to 30 cm distance i.e twice as compared to 30 cm.

Response: The reviewer is right that there are many different factors involved to explain differences between 30-cm and 60-cm testing condition. We agree with the reviewer that the double spray load should certainly be mentioned, perhaps even as the first factor, as this was a testing condition. We have added the following text in the discussion section (L235-L247): ‘To avoid a too low recovery on the membrane filter we doubled the spray load for the 60 cm test condition. This two-fold increase of the source strength also contributed to the difference observed between the 30 and 60 cm test condition. As can be seen in Figure 3 the TIL is higher at the 60 cm condition compared to the 30 cm condition. As the concentration decreases exponentially with distance, the covered fluorescein would be expected to be lower at 60 cm compared to 30 cm even when taking into account the twofold in-crease of load. Finding similar or even higher TIL and a much smaller variance indicates that small droplets (that remain airborne at 60 cm have a relatively higher contribution to the overall TIL performance as compared to the 30 cm condition. There are two reasons why smaller particles are more abundant: large droplets did not reach the 60 cm point because of gravity and at 60 cm the droplets have lost more of their size due to water evaporation. Despite the smaller droplet size, the amount of fluorescein remains the same’.

Reviewer 2 Report

In this paper, the authors report an experimental procedure to test the performance of respiratory protective equipment (RPE) using a custom-built nebulizer with open outlet and a non-volatile fluorescent probe in the water spray. The total inward leakage (TIL) was used as the parameter to judge and compare the performance of two types of RPE, which are FFP2 and surgical masks. Considering the pandemic that we are facing at present, this is a topical subject of research and of interest to a broad readership. The paper is well written and divided in sections. Also the methodology is well described. 

  1. Since the TIL is contributed also by the face seal leakage, the angle of incidence of the jet spray may affect the results? The authors tested the performance pointing to the nose, namely the incident direction was completely perpendicular to the frontal plane of the face. Did the authors consider if two types of RPE may have different leakage from the side if they fit to the face differently? In this case, may jet spray with angle of incidence not normal to the facial plane show different performance of the two RPE?
  2. It is not clear to me the sample preparation for the fluorescence measurement of test samples and standards for calibration. In case of test solutions, the fluorescein on the membrane filter was recovered after extraction by mechanical shaking in Mq. For the calibration, known quantities of fluorescein were applied on the membrane filter surface and then recovered with the same procedure as for testing samples. Am I correct? Please clarify this point.
  3. I suppose that the aliquots for measurements of fluorescence from test samples and calibration samples must be of equal volume. The authors should confirm it and possibly add the volume (in microliter, I presume).
  4. The results of the calibration are not shown. The extremely good linearity of the calibration line given by R2 implies that the yield of recovery of fluorescein was independent on the initial concentration on the membrane filter. Did standard deviation at each concentration from standards small? How many measurements for each concentration with standards? The calculated R2 is 0.98 or 0.99? Usually calibration is obtained from one fitting line from the plot of means of many measurements at different concentrations, which means that only one R2 should be presented.
  5. Why did the authors used Bonferroni correction? For the 30 cm case, they analyzed 4 different groups with only 1 test (i.e., 1 hypothesis based on difference in TIL). Why not ANOVA followed by post-hoc test for comparison between 2 groups? In this case, also at 30 cm the difference between surgical mask and FFP2 mask would probably be significant. Please justify the choice of statistical analysis.
  6. In fig. 3, please add (a) and (b) in the two figures. Moreover, it would be interesting to add the box and whiskers plot of the data for the case of “face shield only” also in Fig. 3(a).
  7. In figure S3 of Supplementary material, please add axis labels.

Author Response

The authors much appreciate the very useful suggestions for clarification and improvement of the manuscript. Below we have responded point-by-point and also suggested some changes to resolve the concerns of the reviewers.

Reviewer-2

In this paper, the authors report an experimental procedure to test the performance of respiratory protective equipment (RPE) using a custom-built nebulizer with open outlet and a non-volatile fluorescent probe in the water spray. The total inward leakage (TIL) was used as the parameter to judge and compare the performance of two types of RPE, which are FFP2 and surgical masks. Considering the pandemic that we are facing at present, this is a topical subject of research and of interest to a broad readership. The paper is well written and divided in sections. Also the methodology is well described.

Response: We appreciate the reassuring words of the reviewer regarding relevance of our work, the quality of the writing and the clarity of the methodology description.

Since the TIL is contributed also by the face seal leakage, the angle of incidence of the jet spray may affect the results? The authors tested the performance pointing to the nose, namely the incident direction was completely perpendicular to the frontal plane of the face. Did the authors consider if two types of RPE may have different leakage from the side if they fit to the face differently? In this case, may jet spray with angle of incidence not normal to the facial plane show different performance of the two RPE?

Response: The reviewer is right to point out that the angle of the jet spray is an important test condition that will likely influence the result. We chose to point the jet spray at the nose to allow a well-standardized and well-aligned test condition for optimal measurement repeatability. In the laboratory setting this is a good solution but as the reviewer indicates, in reality an exposure may be from a variable and different angle. To make the reader aware of this we have added the following to the discussion section (L235-246): ‘It would be useful to consider to add a spray condition at a different angle, e.g. from the side or from below: it is likely in the case of the surgical mask that e.g., duct-shaped pleats formed at the side of the head may enhance the contribution of face-seal leakage.’

It is not clear to me the sample preparation for the fluorescence measurement of test samples and standards for calibration. In case of test solutions, the fluorescein on the membrane filter was recovered after extraction by mechanical shaking in Mq. For the calibration, known quantities of fluorescein were applied on the membrane filter surface and then recovered with the same procedure as for testing samples. Am I correct? Please clarify this point.

Response: Good to verify the reviewer’s interpretation of the current methods description and we are pleased to confirm that this is a correct interpretation of the what we did.

I suppose that the aliquots for measurements of fluorescence from test samples and calibration samples must be of equal volume. The authors should confirm it and possibly add the volume (in microliter, I presume).

Response: We can reassure the reviewer that all fluorescence measurements were done in the same volume of 200 microliter in the wells of the 96-wells plate and have clarified the text in L141.

The results of the calibration are not shown. The extremely good linearity of the calibration line given by R2 implies that the yield of recovery of fluorescein was independent on the initial concentration on the membrane filter. Did standard deviation at each concentration from standards small? How many measurements for each concentration with standards? The calculated R2 is 0.98 or 0.99? Usually calibration is obtained from one fitting line from the plot of means of many measurements at different concentrations, which means that only one R2 should be presented.

Response: We appreciate the reviewer’s interest to better clarify the preparation of standards for the calibration. The reviewer is right that we used the same procedure for the standards and the samples. We also appreciate the reviewer’s critical appraisal of the way we present the calibration results. The day-to-day variability of the calibration has been added as a relative standard deviation. We have revised the description in the methods section L153-L156) to clarify this: ‘On every day RPEs were tested we prepared a new calibration curve.  The mean R2 corresponds to 0.98 with variability in the slope factor3.2 %, calculated as relative standard deviation based on 4 calibration curves. Filter and solution blanks and the calibration standards were analysed in the same run (on the same 96 wells plate).

Why did the authors used Bonferroni correction? For the 30 cm case, they analyzed 4 different groups with only 1 test (i.e., 1 hypothesis based on difference in TIL). Why not ANOVA followed by post-hoc test for comparison between 2 groups? In this case, also at 30 cm the difference between surgical mask and FFP2 mask would probably be significant. Please justify the choice of statistical analysis.

Response: We appreciate the critical remarks regarding the statistical analysis. We thank the reviewer for the suggestion. We aimed to have a rather conservative multiple testing procedure given the limited sample size, hence the Bonferroni correction. Other choices are now post-hoc and could be considered opportunistic. Leaving that aside for the moment, if a post-hoc Tukey test would be performed, the q-value equal to (TIL(FFP2) – TIL (SM)) / (SE(TIL(FFP2)-TIL(SM)) = 2.19 would not be statistically significant at 0.05 two-sided (52 degrees of freedom), as its value q=2.19 does not exceed the critical q-value of 2.65 for all pairwise differences between 4 groups. For considering 3 groups, the critical value is 2.41, so the same conclusion would hold. It is only if we had prespecified to only test for the difference between FFP2 and surgical mask that this difference would have been statistically significant. 

Also, we believe that the amount or lack of overlap is of more practical important when comparing the distributions than the difference in means.

In fig. 3, please add (a) and (b) in the two figures. Moreover, it would be interesting to add the box and whiskers plot of the data for the case of “face shield only” also in Fig. 3(a).

Response: We have added the labels (a) and (b) to Figure 3. We agree that it is useful to also present the data on the face-shield-only test at 30 cm in Figure 3a and have added this box plot with whiskers in Figure 3(a).

In figure S3 of Supplementary material, please add axis labels.

Response: In Figure S3 we added the unit ‘µm’ on the X. On Y we presented the cumulative distribution based on frequency data (presented on Y) that does not have a dimension. We have added ‘[-]’ as a label on Y to indicate more clearly that there is no unit on Y.

Round 2

Reviewer 2 Report

The authors answered properly to my comments. 

In Figure S3 of Supplementary material I suggest to label the x axis as "Particle size (um)" and y axis as "Cumulative particle fraction".